# Back2Future: Leveraging Backfill Dynamics for Improving Real-time Predictions in Future

**Harshavardhan Kamarthi, Alexander Rodríguez, B. Aditya Prakash**
College of Computing
Georgia Institute of Technology
`{harsha.pk,arodriguezc,badityap}@gatech.edu`

## Abstract

For real-time forecasting in domains like public health and macroeconomics, data collection is a non-trivial and demanding task. Often after being initially released, it undergoes several revisions later (maybe due to human or technical constraints) - as a result, it may take weeks until the data reaches a stable value. This so-called 'backfill' phenomenon and its effect on model performance have been barely addressed in the prior literature. In this paper, we introduce the multi-variate backfill problem using COVID-19 as the motivating example. We construct a detailed dataset composed of relevant signals over the past year of the pandemic. We then systematically characterize several patterns in backfill dynamics and leverage our observations for formulating a novel problem and neural framework, Back2Future, that aims to refines a given model's predictions in real-time. Our extensive experiments demonstrate that our method refines the performance of diverse set of top models for COVID-19 forecasting and GDP growth forecasting. Specifically, we show that Back2Future refined top COVID-19 models by 6.65% to 11.24% and yield 18% improvement over non-trivial baselines. In addition, we show that our model improves model evaluation too; hence policy-makers can better understand the true accuracy of forecasting models in real-time.

## 1 Introduction

The current COVID-19 pandemic has challenged our response capabilities to large disruptive events, affecting the health and economy of millions of people. A major tool in our response has been forecasting epidemic trajectories which enabled policymakers to plan interventions (Holmdahl & Buckee, 2020). Broadly two classes of approaches have been devised: traditional mechanistic epidemiological models (Shaman & Karspeck, 2012; Zhang et al., 2017), and the fairly newer statistical approaches (Brooks et al., 2018; Adhikari et al., 2019; Osthus et al., 2019b) including deep learning models (Adhikari et al., 2019; Panagopoulos et al., 2021; Rodríguez et al., 2021a), which have become among the top-performing ones for multiple forecasting tasks (Reich et al., 2019). These models also use newer digital indicators like search queries (Ginsberg et al., 2009; Yang et al., 2015) and social media (Culotta, 2010). Epidemic forecasting is still a challenging enterprise (Metcalf & Lessler, 2017; Biggerstaff et al., 2018) because it is affected by weather, mobility, strains, and others.

However, *real-time* forecasting also brings new challenges. As noted in multiple CDC real-time forecasting initiatives for diseases like flu (Osthus et al., 2019a) and COVID-19 (Cramer et al., 2021), as well as in macroeconomics (Clements & Galvão, 2019; Aguiar, 2015) the initially released public health data is revised many times after and is known as the 'backfill' phenomenon.The various factors that affect backfill are multiple and complex, ranging from surveillance resources to human factors like coordination between health institutes and government organizations within and across regions (Chakraborty et al., 2018; Reich et al., 2019; Altieri et al., 2021; Stierholz, 2017).

While previous works have addressed anomalies (Liu et al., 2017), missing data (Yin et al., 2020), and data delays (Žliobaite, 2010) in general time-series problems, the backfill problem has not been addressed. In contrast, the topic of revisions has not received as much attention, with few exceptions. For example in epidemic forecasting, a few papers have either (a) mentioned about the 'backfill problem' and its effects on performance (Chakraborty et al., 2018; Rodríguez et al., 2021b; Altieri

et al., 2021; Rangarajan et al., 2019) and evaluation (Reich et al., 2019); or (b) proposed to address the problem via simple models like linear regression (Chakraborty et al., 2014) or 'backcasting' (Brooks et al., 2018) the observed targets. The related problem of *nowcasting* involves prediction of revised stable value of current week's target from sequence of right-truncated past values [Aditya: cite one paper ]. Prior works have used data assimilation and sensor fusion from a readily available stable set of features to refine unrevised features for accurate nowcasting (Farrow, 2016; Osthus et al., 2019a). However, most methods focus only on revisions in the *target* and typically study in the context of influenza forecasting, which is substantially less noisy and more regular than the novel COVID-19 pandemic or assume access to stable values for some features which is not the case for COVID-19. In economics, Clements & Galvão (2019) surveys several domain-specific (Carriero et al., 2015) or essentially linear techniques for data revision/correction behavior of several macroeconomic indicators (Croushore, 2011).

Motivated from above, we study the challenging problem of multi-variate backfill for *both* features and targets. We go further beyond prior work and also show how to leverage our insights towards a general neural framework to *improve* model predictions and performance evaluation (i.e. *rectification* of current target from the evaluator's perspective). Our specific contributions are the following:

• **Multi-variate backfill problem:** We introduce the multi-variate backfill problem using real-time epidemiological forecasting as the primary motivating example. In this challenging setting, which generalizes (the limited) prior work, the forecast targets, as well as exogenous features, are subject to retrospective revision. Using a carefully collected diverse dataset for COVID-19 forecasting for the past year, we discover several patterns in backfill dynamics, show that there is a significant difference in real-time and revised feature measurements, and highlight the negative effects of using unrevised features for incidence forecasting in different models both for model performance and evaluation. Building on our empirical observations, we formulate the problem BFRP, which aims to 'correct' given model predictions to achieve better performance on eventual fully revised data.

• **Spatial and Feature level backfill modeling to refine model predictions:** Motivated by the patterns in revision and observations from our empirical study, we propose a deep-learning model Back2Future (B2F) to model backfill revision patterns and derive latent encodings for features. B2F combines Graph Convolutional Networks that capture sparse, cross-feature, and cross-regional backfill dynamics similarity and deep sequential models that capture temporal dynamics of each features' backfill dynamics across time. The latent representation of all features is used along with the history of the model's predictions to improve diverse classes of models trained on real-time targets, to predict targets closer to revised ground truth values. Our technique can be used as a 'wrapper' to improve model performance of any forecasting model (mechanistic/statistical).

• **Refined top models' predictions and improved model evaluation:** We perform an extensive empirical evaluation to show that incorporating backfill dynamics through B2F consistently improves the performance of diverse classes of top-performing COVID-19 forecasting models (from the CDC COVID-19 Forecast Hub, including the top-performing official ensemble) significantly. B2F also enables forecast evaluators and policy-makers better evaluate the 'eventual' true accuracy of participating models (against revised ground truth). This allows the model evaluators to quickly estimate models that perform better w.r.t revised stable targets instead of potentially misleading current targets. Our methodology can also be further adapted for nowcasting and other general time-series forecasting problems. We also show the generalizability of our framework and model B2F to other domains by significantly improving predictions of non-trivial baselines for US National GDP forecasting (Marcellino, 2008).

## 2   NATURE OF BACKFILL DYNAMICS

In this section, we study important properties of the revision dynamics of our signals. We introduce some concepts and definitions to aid in the understanding of our empirical observations and method.

**Real-time forecasting.** We are given a set of signals $\mathcal{F} = \text{Reg} \times \text{Feat}$, where $\text{Reg}$ is the set of all regions (where we want to forecast) and set $\text{Feat}$ contains our features and forecasting target(s) for each region. At prediction week $t$, $x_{i,1:t}^{(t)}$ is a time series from 1 to $t$ for feature $i$, and the set of all signals results in the multi-variate time series $\mathcal{X}_{1:t}^{(t)1}$. Similarly, $\mathcal{Y}_{1:t}^{(t)}$ is the forecasting target(s) time

---

[1] In practice, delays are possible too, i.e, at week $t$, we have data for some feature $i$ only until $t - \delta_i$. All our results incorporate these situations. We defer the minor needed notational extensions to Appendix for clarity.

series. Further, let's call all data available at time $t$, $\mathcal{D}_{1:t}^{(t)} = \{\mathcal{X}_{1:t}^{(t)}, \mathcal{Y}_{1:t}^{(t)}\}$ as *real-time sequence*. For clarity we refer to 'signal' $i \in \mathcal{F}$ as a sequence of either a feature or a target, and denote it as $d_{i,1:t}^{(t)}$. Thus, at prediction week $t$, the real-time forecasting problem is: *Given $\mathcal{D}_{1:t}^{(t)}$, predict next $k$ values of forecasting target(s), i.e. $\hat{y}_{t+1:t+k}$.* Typically for CDC settings and this paper, our time unit is week, $k = 4$ (up to 4 weeks ahead) and our target is COVID-19 mortality incidence (Deaths).

**Revisions.** Data revisions ('backfill') are common. At prediction week $t + 1$, the real-time sequence $\mathcal{D}_{1:t+1}^{(t+1)}$ is available. In addition to the length of the sequences increasing by one (new data point), values of $\mathcal{D}_{1:t+1}^{(t+1)}$ already in $\mathcal{D}_{1:t}^{(t)}$ may be revised i.e., $\mathcal{D}_{1:t}^{(t)} \neq \mathcal{D}_{1:t}^{(t+1)}$. Note that previous work has studied backfill limited to $\mathcal{Y}^{(t)}$, while we address it in both $\mathcal{X}^{(t)}$ and $\mathcal{Y}^{(t)}$. Also, note that the data in the backfill is the same used for real-time forecasting, but just seen from a different perspective.

**Backfill sequences:** Another useful way we propose to look at backfill is by focusing on revisions of a single value. Let's focus on value of signal $i$ at an *observation week* $t'$. For this observation week, the value of the signal can be revised at any $t > t'$, which induces a sequence of revisions. We refer to *revision week* $r \geq 0$ as the relative amount of time that has passed since the observation week $t'$.

**Defn. 1.** *(Backfill Sequence* BSEQ*)* For signal $i$ and observation week $t'$, its backfill sequence is $\text{BSEQ}(i, t') = \langle d_{i,t'}^{(t')}, d_{i,t'}^{(t'+1)}, \ldots, d_{i,t'}^{(\infty)} \rangle$, where $d_{i,t'}^{(t')}$ is the underline{initial value} of the signal and $d_{i,t'}^{(\infty)}$ is the final/stable value of the signal.

**Defn. 2.** *(Backfill Error* BERR*)* For revision week $r$ of a backfill sequence, the backfill error is $\text{BERR}(r, i, t') = |d_{i,t'}^{(t'+r)} - d_{i,t'}^{(\infty)}| \,/\, |d_{i,t'}^{(\infty)}|$.

**Defn. 3.** *(Stability time* STIME*) of a backfill sequence* BSEQ *is the revision week $r^*$ that is the minimum $r$ for which the backfill error* BERR $< \epsilon$ *for all $r > r^*$, i.e., the time when* BSEQ *stabilizes.*

*Note:* We ensured that BSEQ length is at least 7, and found that in our dataset most signals stabilize before $r = 20$. For $d_{i,t'}^{(\infty)}$, we use $d_{i,t'}^{(t_f)}$, at the final week $t_f$ in our revisions dataset. In case we do not find BERR $< \epsilon$ in any BSEQ, we set STIME to the length of that BSEQ. We use $\epsilon = 0.05$. **Example:** For BSEQ $\{223, 236, 236, 404, \ldots, 404\}$, BERR for third week is $\frac{|236-404|}{404} = 0.41$ and STIME is 4.

## 2.1 DATASET DESCRIPTION

We collected and pre-processed important publicly available signals from a variety of trusted sources that are relevant to COVID-19 forecasting to form the COVID-19 *Surveillance Dataset* (*CoVDS*). See Table 1 for the list of 20 features (|Feat| = 21, including Deaths). We collected revised features every week from April 2020 to July 2021. Our analysis covers 30 observation weeks from June 2020 to December 2020 (to ensure all our backfill sequences are of length at least 7) for all |Reg| = 50 US states. The rest of the *unseen* data from Jan 2021 to July 2021 is used strictly for evaluation.

**Patient line-list:** traditional surveillance signals used in epidemiological models (Chakraborty et al., 2014; Brooks et al., 2018) derived from line-list records e.g. hospitalizations from CDC (CDC, 2020), positive cases, ICU admissions from COVID Tracking (COVID-Tracking, 2020). **Testing:** measure changes in testing from CDC and COVID-Tracking used by Rodríguez et al. (2021b). **Mobility:** quantify change in people's movement to several point of interests (POIs); derived from mobility reports released by Google (2020); Apple (2020). **Exposure:** digital signal measuring closeness between people at POIs,(Chevalier et al., 2021) **Social Survey:** used by (Wang et al., 2020; Rodríguez et al., 2021b) CMU/Facebook Symptom Survey Data contains self-reported responses about COVID-19 symptoms.

Table 1: List of features in our *CoVDS*

| Type | Features |
|---|---|
| Patient Line-List | ERVisits, HospRate, +veInc, HospInc, Recovered, onVentilator, inICU |
| Testing | TestResultsInc, -veInc, Facilities |
| Mobility | RetailRec, Grocery, Parks, Transit, WorkSpace, Resident, AppleMob |
| Exposure | DexA |
| Social Survey | FbCLI, FbWiLi |

## 2.2 OBSERVATIONS

We first study different facets of the significance of backfill in *CoVDS*. Using our definitions, we generate a backfill sequence for every combination of signal, observation week, and region (not all signals are available for all regions). In total, we generate *more than* $30,000$ backfill sequences.

**Backfill error BERR is significant.** We computed BERR for the initial values, i.e., $\text{BERR}(r = 0, i, t')$, for all signals $i$ and observation weeks $t'$.

**Obs. 1.** *(BERR across signals and regions) Compute the average BERR for each signal; the median of all these averages is 32%, i.e. at least half of all signals are corrected by 32% of their initial value. Similarly in at least half of the regions the signal corrections are 280% of their initial value.*

We also found large variation of BERR. For features (Figure 1a), compare avg. $\text{BERR} = 1743\%$ of five most corrected features with 1.6% of the five least corrected features. Also, in contrast to related work that focuses on traditional surveillance data Yang et al. (2015), perhaps unexpectedly, we found that digital indicators also have a significant BERR (average of 108%). For regions (see Figure 1b), compare 1594% of the five most corrected regions with 38% of the five least corrected regions.

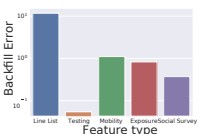 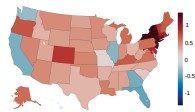 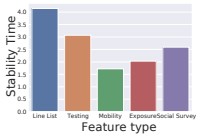 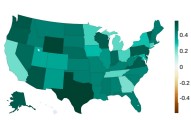

(a) BERR per feat. type    (b) BERR per region    (c) STIME per feat. type    (d) STIME per region

Figure 1: BERR *and* STIME *across feature type and regions, heat maps are* log *scaled.*

**Stability time STIME is significant.** A similar analysis for STIME found significant variation across signals (from 1 weeks for to 21 weeks for COVID-19, see Figure 1c for STIME across feature types) and regions (from 1.55 weeks for GA to 3.83 weeks for TX, see Figure 1d). This also impacts our target, thus, actual accuracy is not readily available which undermines real-time evaluation and decision making.

**Obs. 2.** *(STIME of features and target) Compute the average STIME for each signal; the average of all these averages for features is around 4 weeks and for our target* Deaths *is around 3 weeks, i.e. on average, it takes over 3 weeks to reach the stable values of features.*

**Backfill sequence BSEQ patterns.** There is significant similarity among BSEQs. We cluster BSEQs via K-means using Dynamic Time Warping (DTW) as pair-wise distance (as DTW can handle sequences of varying magnitude and length). We found five *canonical* categories of behaviors (see Figure 2), each of size roughly 11.58% of all BSEQs. Also, each cluster is not defined only by signal

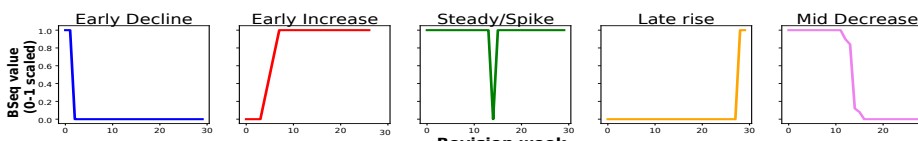

Figure 2: *Centroid* BSEQ *of each cluster, scaled between* $[0, 1]$, *showing canonical backfill behaviors*

nor region. Hence there is a non-trivial similarity across both signals and regions.

**Obs. 3.** *(BSEQ similarity and variety) Five canonical behaviors were observed in our backfill sequences (Figure 2). No cluster has over 21% of BSEQs from the same region, and no cluster has over 14% of BSEQs from the same signal.*

**Model performance vs BERR.** To study the relationship between model performance (via Mean Absolute Error MAE of a prediction) and BERR, we use REVDIFFMAE: the difference between MAE computed against *real-time* target value and one against the *stable* target value. We analyze the top-performing real-time forecasting models as per the comprehensive evaluation of all models in COVID-19 Forecast Hub (Cramer

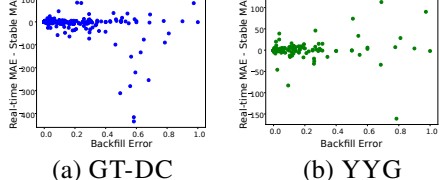

(a) GT-DC      (b) YYG

Figure 3: BERR *vs model* REVDIFFMAE.

et al., 2021). YYG and UMASS-MB are mechanistic while CMU-TS and GT-DC are statistical models. The top performing ENSEMBLE is composed of all contributing models to the hub. We expect a well-trained real-time model will have higher REVDIFFMAE with larger BERR in its target (Reich et al., 2019). However, we found that higher BERR does not necessarily mean worse performance. See Figure 3—YYG has even better performance with more revisions. This may be due to the more complex backfill activity/dependencies in COVID in comparison to the more regular seasonal flu.

**Obs. 4.** *(Model performance and backfill) Relation between BERR and REVDIFFMAE can be non-monotonous and positively or negatively correlated depending on model and signal.*

**Real-time target values to measure model performance**: Since targets undergo revisions (5% BERR on average), we study how this BERR affects the real-time evaluation of models. From Figure 4, we see that the scores are not similar with real-time scores over-estimating model accuracy. The average difference in scores is positive which implies that evaluators would overestimate models' forecasting ability.

**Obs. 5.** *MAE evaluated at real-time overestimates model performance by 9.6 on average, with the maximum for TX at 22.63.*

## 3   REFINING THE FUTURE VIA B2F

Our observations naturally motivate improving training and evaluation aspects of *real-time* forecasting by leveraging revision information. Thus, we propose the following two problems. Let predictions of model $M$ for week $t + k$ be $y(M, k)_t$. Since models are trained on *real-time* targets, $y(M, k)_t$ is the model's estimate of target $y_{t+k}^{(t+k)}$.

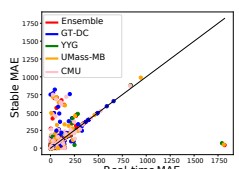

Figure 4: *Real-time vs stable MAE.*

$k$**-week ahead Backfill Refinement Problem, BFRP$_k(M)$:** At prediction week $t$, we are given a *revision dataset* $\{\mathcal{D}_{1:t'}^{(t')}\}_{t' \leq t}$, which includes our target $\mathcal{Y}_{1:t}^{(t)}$. For a model $M$ trained on *real-time* targets, given history of model's predictions till last week $\langle y(M, k)_1, \ldots y(M, k)_{t-1} \rangle$ and prediction for current week $y(M, k)_t$, our goal is to refine $y(M, k)_t$ to better estimate the *stable* target $y_{t+k}^{(t_f)}$, i.e. the 'future' of our target value at $t + k$.

**Leaderboad Refinement problem, LBRP:** At each week $t$, evaluators are given a current estimate of our target $y_t^{(t)}$ and forecasts of models submitted on week $t - k$. Our goal is to refine $y_t^{(t)}$ to $\hat{y}_t$, a better estimate of $y_t^{(t_f)}$, so that using $\hat{y}_t$ as a surrogate for $y_t^{(t_f)}$ to evaluate predictions of models provides a better indicator of their actual performance (i.e., we obtain a refined leaderboard of models). Since LBRP aims to refine the current estimate of the target, it is closely related to the nowcasting problem. LBRP *is also a special case of* BFRP: Assume a hypothetical model $M_{eval}$ whose predictions are real-time ground truth, i.e. $y(M_{eval}, 0)_t = y_t^{(t)}, \forall t$. Then, refining $M_{eval}$ is equivalent to refining $y_t^{(t)}$ to better estimate $y_t^{(t_f)}$ which leads to solving LBRP.

**Overview:** We leverage observations from Section 2 to derive Back2Future (B2F), a deep-learning model that uses revision information from BSEQ to refine predictions. Obs. 1 and 2 show that real-time values of signals are poor estimates of stable values. Therefore, we leverage patterns in BSEQ of past signals and exploit cross-signal similarities (Obs. 3) to extract information from BSEQs. We also consider that the relation of models' forecasts to BERR of targets is complex (Obs. 4 and 5) to refine their predictions. B2F combines these ideas through its four modules: • GRAPHGEN: Generates a signal graph (where each node maps to a signal in Reg × Feat) whose edges are based on BSEQ similarities. • BSEQENC: Leverages the signal graph as well as temporal dynamics of BSEQs to learn a latent representation of BSEQs using a Recurrent Graph Neural Network. • MODELPREDENC: Encodes the history of the model's predictions, the real-time value of the target, and past revisions of the target through a recurrent neural network. • REFINER: Combines encodings from BSEQENC and MODELPREDENC to predict the correction to model's real-time prediction.

In contrast to previous works that studies target BERR (Reich et al., 2019), we simultaneously model all BSEQ available till current week $t$ using spatial and signal similarities in the temporal dynamics of BSEQ. Recent works that attempt to model spatial relations for COVID19 forecasting need explicitly structural data (like cross-region mobility) (Panagopoulos et al., 2020) to generate a graph or use attention over temporal patterns of regions' death trends. B2F, in contrast, directly models the structural information of signal graph (containing features from each region) using BSEQ similarities. Thus, we first generate useful latent representations for each signal based on BSEQ revision information of that feature as well as features that have shown similar revision patterns in the past. Due to the large number of signals that cover all regions, we cannot model the relations between every pair using fully connected modules or attention similar to (Jin et al., 2020). Therefore, we first construct a sparse graph between signals based on past BSEQ similarities. Then we inject this similarity information using Graph Convolutional Networks (GCNs) and combine it with deep sequential models to model temporal dynamics of BSEQ of each signal while combining information from BSEQ s of signals in the neighborhood of the graph. Further,

we use these latent representations and leverage the history of a model $M$'s predictions to refine its prediction. Thus, B2F solves $\text{BFRP}_k(M)$ assuming $M$ is a black box, accessing only its past forecasts. Our training process, that involves pre-training on model-agnostic auxiliary task, greatly improves training time for refining any given model $M$. The full pipeline of B2F is also shown in Figure 5. Next, we describe each of the components of B2F in detail. For the rest of this section, we will assume that we are forecasting $k$ weeks ahead given data till current week $t$.

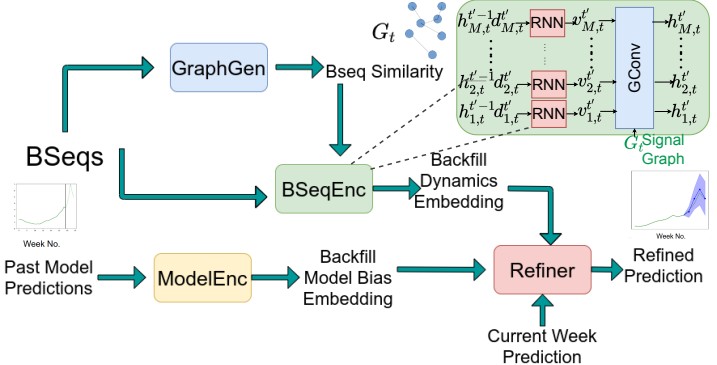

Figure 5: B2F *pipeline with all components*

**GRAPHGEN** generates an undirected signal graph $G_t = (V, E_t)$ whose edges represent similarity in BSEQs between signals, where vertices $V = \mathcal{F} = \text{Reg} \times \text{Feat}$. We measure similarity using DTW distance due to reasons described in Section 2. GRAPHGEN leverages the similarities across BSEQ patterns irrespective of the exact nature of canonical behaviors which may vary across domains. We compute the sum of DTW distances of BSEQs for each pair of nodes summed over $t' \in \{1, 2, \ldots, t-5\}$. We threshold $t'$ till $t-5$ to make the BSEQs to be of reasonable length (at least 5) to capture temporal similarity without discounting too many BSEQs. Top $\tau$ node pairs with lowest total DTW distance are assigned an edge.

**BSEQENC.** While we can model backfill sequences for each signal independently using a recurrent neural network, this doesn't capture the behavioral similarity of BSEQ across signals. Using a fully-connected recurrent neural network that considers all possible interactions between signals also may not learn from the similarity information due to the sheer number of signals ($50 \times 21 = 1050$) while greatly increasing the parameters of the model. Thus, we utilize the structural prior of graph $G_t$ generated by GRAPHGEN and train an autoregressive model BSEQENC which consists of graph recurrent neural network to encode a latent representation for each of backfill sequence in $B_t = \{\text{BSEQ}(i, t) : i \in \mathcal{F}\}$. At week $t$, BSEQENC is first pre-trained and then it is fine-tuned for a specific model $M$ (more details later in this section).

Our encoding process is in Figure 5. Let $\text{BSEQ}_{t'+r}(i, t')$ be first $r + 1$ values of $\text{BSEQ}(i, t')$ (till week $t' + r$). For a past week $t'$ and revision week $r$, we denote $h_{i,t'}^{(t'_r)} \in \text{R}^m$ to be the latent encoding of $\text{BSEQ}_{t'_r}(i, t')$ where $t'_r = t' + r$ and $t' \leq t'_r \leq t$. We initialize $h_{i,t'}^{(0)}$ for any observation week $t'$ to be a learnable parameter $h_i^{(0)} \in \text{R}^m$ specific to signal $i$. For each week $t'_r$ we combine latent encoding $h_{i,t'}^{(t'_r-1)}$ and signal value $d_{i,t'}^{(t'_r)}$ using a GRU (Gated Recurrent Unit) (Cho et al., 2014) cell to get the intermediate embedding $v_{i,t'}^{(t'_r)}$. Then, we leverage the signal graph $G_t$ and pass the embeddings $\{v_{i,t'}^{(t'_r)} : i \in \mathcal{F}\}$ through a Graph Convolutional layer (Kipf & Welling, 2016) to get $h_{i,t'}^{(t'_r)}$:

$$v_{i,t'}^{(t'_r)} = \text{GRU}_{\text{BE}}(d_{i,t'}^{(t'_r)}, h_{i,t'}^{(t'_r-1)}), \qquad \{h_{i,t'}^{(t'_r)}\}_{i \in \mathcal{F}} = \text{GConv}(G_t, \{v_{i,t'}^{(t'_r)}\}_{i \in \mathcal{F}}). \tag{1}$$

Thus, $h_{i,t'}^{(t'_r)}$ contains information from $\text{BSEQ}_{t'_r}(i, t')$ and structural priors from $G_t$. Using $h_{i,t'}^{(t'_r)}$, BSEQENC predicts the value $d_{i,t'}^{(t'_r+1)}$ by passing through a 2-layer feed-forward network $\text{FFN}_i$: $\hat{d}_{i,t'}^{(t'_r+1)} = \text{FFN}_i(h_{i,t'}^{(t'_r)})$. During inference, we only have access to real-time values of signals for the current week. We autoregressively predict $h_{i,t}^{(t+l)}$ for each signal by initially passing $\{d_{i,t}^{(t)}\}_{i \in \mathcal{F}}$ through BSEQENC and using the output $\{\hat{d}_{i,t}^{(t+1)}\}_{i \in \mathcal{F}}$ as input for BSEQENC. Iterating this $l$ times we get $\{h_{i,t}^{(t+l)}\}_{i \in \mathcal{F}}$ along with $\{\hat{d}_{i,t}^{(t+l)}\}_{i \in \mathcal{F}}$ where $l$ is a hyperparameter.

**MODELPREDENC.** To learn from history of a model's predictions and its relation to target revisions, MODELPREDENC encodes the history of model's predictions, previous real-time targets, and revised (up to current week) targets using a Recurrent Neural Network. Given a model $M$, for each observation week $t' \in \{1, 2, \ldots, t-1-k\}$, we concatenate the model's predictions $y(M, k)_{t'}$, real-time target

$y_{t'+k}^{(t'+k)}$ as seen on observation week $t'$ and revised target $y_{t'+k}^{(t)}$ as $C_{t'}^t = y(M,k)_{t'} \oplus y_{t'+k}^{(t'+k)} \oplus y_{t'+k}^{(t)}$, where $\oplus$ is the concatenation operator. A GRU is used to encode the sequence $\{C_1^{(t)}, \ldots, C_{t-1-k}^{(t)}\}$:

$$\{z_1^{(t)}, \ldots, z_{t-1-k}^{(t)}\} = \text{GRU}_{\text{ME}}(\{C_1^{(t)}, \ldots, C_{t-1-k}^{(t)}\}) \tag{2}$$

**REFINER.** It leverages the information from above three modules of B2F to refine model $M$'s prediction for current week $y(M,k)_t$. Specifically, it receives the latent encodings of signals $\{h_{i,t}^{(t+l)}\}_{i \in \mathcal{F}}$ from BSEQENC, $z_{t-k-1}^{(t)}$ from MODELPREDENC, and the model's prediction $y(M,k)_t$ for week $t$.

BSEQ encoding from different signals may have variable impact on refining the signal since a few signals may not very useful for current week's forecast (e.g., small revisions in mobility signals may not be important in some weeks). Moreover, because different models use signals from *CoVDS* differently, we may need to focus on some signals over others to refine its prediction. Therefore, we first take attention over BSEQ encodings from all signals $\{h_{i,t}^{(t+l)}\}_{i \in \mathcal{F}}$ w.r.t $y(M,k)_t$. We use multiplicative attention mechanism with parameter $w \in \mathbb{R}^m$ based on Vaswani et al. (2017):

$$\alpha_i = \text{softmax}_i(y(M,k)_t w_h^T h_{i,t}^{t+l}), \qquad \bar{h}^{(t)} = \sum_{i \in \mathcal{F}} \left( \alpha_i h_{i,t}^{t+l} \right). \tag{3}$$

Finally we combine $\bar{h}^{(t)}$ and $z_{t-k-1}^{(t)}$ through a two layer feed-forward layer $\text{FNN}_{\text{RF}}$ which outputs a 1-dim value followed by $\tanh$ activation to get the correction $\gamma_t \in [-1, 1]$ i.e., $\gamma_t = \tanh(\text{FFN}_{\text{RF}}(\bar{h}^{(t)} \oplus z_{t-k-1}^{(t)}))$. Finally, the refined prediction is $y^*(M,K)_t = (\gamma+1)y(M,K)_t$. Note that we limit the correction by B2F by at most the magnitude of model's prediction because the average BERR of targets is $4.9\%$ and less than $0.6\%$ of them have BERR over 1. Therefore, we limit the refinement of prediction to this range.

**Training:** There are two steps of training involved for B2F: 1) model agnostic autoregressive BSEQ prediction task to pre-train BSEQENC; 2) model-specific training for BFRP.

*Autoregressive* BSEQ *prediction:* Pre-training on auxiliary tasks to improve the quality of latent embedding is a well-known technique for deep learning methods (Devlin et al., 2019; Radford et al., 2018). We pre-train BSEQENC to predict the next values of backfill sequences $\{x_{t',i}^{(t'_r+1)}\}_{i \in \mathcal{F}}$. Note that we only use BSEQ sequences $\{\text{BSEQ}_t(t',i)\}_{i \in \mathcal{F}, t' < t}$ available till current week $t$ for training BSEQENC. The training procedure in itself is similar to Seq2Seq prediction problems (Sutskever et al., 2014) where for initial epochs we use the ground truth inputs at each step (teacher forcing) and then transition to using output predictions of previous time step by the recurrent module as input to next time step. Once we pre-train BSEQENC, we can use it for BFRP as well as LBRP for current week $t$ *for any model $M$*. Fine-tuning usually takes less than half the epochs required for pre-training enabling quick refinement of multiple models in parallel.

*Model specific end-to-end training:* Given the pre-trained BSEQENC, we train, end-to-end, the parameters of all modules of B2F. The training set consists of past model predictions $\langle y(M,k)_1, y(M,k)_2, \ldots y(M,k)_{t-1} \rangle$ and backfill sequences $\{\text{BSEQ}_t(t',i)\}_{i \in \mathcal{F}, t' \leq t}$. For datapoint $y(M,k)_{t'}$ of week $t' < t-k$, we input backfill sequences of signals whose observation week is $t'$ into BSEQENC to get latent encodings $\{h_{t',i}^{(t)}\}_{i \in \mathcal{F}}$. We also derive $z_{t'}^{(t)}$ from MODELPREDENC and finally REFINER ingests $z_{t'}^{(t)}$, $\{h_{t',i}^t\}_{i \in \mathcal{F}}$ and $y(M,k)_{t'}$ to get $\gamma_{t'}$. Overall, we optimize the loss function: $\mathcal{L}^{(t)} = \sum_{i=1}^{t-k-1} \left( \gamma_{t'} y(M,k)_{t'} - y_{t'+k}^{(t)} \right)^2$. Following real-time forecasting, we train B2F each week from scratch (including pre-training). Throughout training and forecasting for week $t$, we use $G_t$ as input to BSEQENC since it captures average similarities in BSEQs till current week $t$.

## 4 BACK2FUTURE EXPERIMENTAL RESULTS

In this section, we describe a detailed empirical study to evaluate the effectiveness of our framework B2F. All experiments were run in an Intel i7 4.8 GHz CPU with Nvidia Tesla A4 GPU. The model typically takes around 1 hour to train for all regions. The appendix contains additional details (all hyperparameters and results for June-Dec 2020 and $k = 1, 3$ and GDP forecasting). We also release the code and datasets at www.github.com/AdityaLab/Back2Future.

**Setup:** We perform real-time forecasting of COVID-19 related mortality (Deaths) for 50 US states. We leveraged observations (Section 2) from BSEQ for period June 2020 - Dec. 2020 to design B2F.

We tuned the model hyperparameters using data from June 2020 to Aug. 2020 and tested it on the rest of dataset including completely *unseen data* from Jan. 2021 to June 2021. For each week $t$, we train the model using the *CoVDS* dataset available till the current week $t$ (including BSEQs for all signals revised till $t$) for training. As described in Section 3, for each week, we first pre-train BSEQENC on BSEQ data and then train all components of B2F for each model we aim to refine. Then, we predict the forecasts `Deaths` $y^*(M, k)_t$ for each model $M$. Similarly we also evaluated B2F for real-time GDP forecasting task with detailed results in Appendix. We observed that setting hyperparameter $\tau = c|\mathcal{F}|$ where $c \in \{2, 3, 4, 5\}$ provided best results. Note that $\tau$ influences the sparsity of the graph as well as the efficiency of the model since sparser graphs lead to fast inference across GConv layers. We also found setting $l = 5$ provided the best performance.

**Evaluation:** Refined prediction $y^*(M, K)_t$ are evaluated against the most revised version of the target $y_{t'+k}^{(t_f)}$, where $t_f$ is second week of Feb 2021. For evaluation, we use standard metrics in this domain Reich et al. (2019); Adhikari et al. (2019). Let absolute error of prediction $e(M, k)_t = |y_{t+k}^{(t_f)} - y^*(M, K)_t|$ for a week $t$ and model $M$. We use (a) Mean Absolute Error $\text{MAE}(M) = \frac{1}{T'} \sum_{i=1}^{T'} | e(M, k)_t |$ and (b) Mean Absolute Percentage Error $\text{MAPE} = \frac{1}{T'} \sum_{i=1}^{T'} e(M, k)_t / | y_{t+k}^{(t_f)} |$.

**Candidate models:** We focus on refining/rectifying the top models from the COVID-19 Forecast Hub described in Section 2; these represent different variety of statistical and mechanistic models.

Table 2: B2F *consistently refines all models. % improvements in MAE and MAPE scores averaged over all regions from Jan 2021 to June 2021*

| Cand. Model | Refining Model | k=2 | | k=4 | |
|---|---|---|---|---|---|
| | | MAE | MAPE | MAE | MAPE |
| ENSEMBLE | FFN | -0.35 ± 0.11 | -0.12 ± 0.22 | 0.87 ± 0.64 | 0.77 ± 0.14 |
| | B2F-MB | -2.23 ± 0.82 | -1.57 ± 0.65 | -2.19 ± 0.35 | -2.85 ± 0.53 |
| | B2F-NOGRAPH | -1.45 ± 0.14 | -2.73 ± 0.35 | -5.72 ± 0.21 | -6.72 ± 0.82 |
| | B2F-BSEQ | 1.42 ± 0.60 | 0.37 ± 0.75 | 0.74 ± 0.36 | 0.44 ± 0.07 |
| | Back2Future | **5.25 ± 0.13** | **4.39 ± 0.62** | **4.41 ± 0.57** | **3.15 ± 0.57** |
| GT-DC | FFN | -2.42 ± 0.22 | -1.51 ± 0.90 | -1.54 ± 0.57 | -0.48 ± 0.43 |
| | B2F-MB | -3.02 ± 0.40 | -3.41 ± 0.16 | -2.91 ± 0.29 | -3.22 ± 0.74 |
| | B2F-NOGRAPH | 2.24 ± 0.37 | 3.51 ± 0.21 | 1.93 ± 0.39 | 0.78 ± 0.37 |
| | B2F-BSEQ | 2.13 ± 0.12 | 3.84 ± 0.78 | 1.08 ± 0.23 | 2.33 ± 0.97 |
| | Back2Future | **10.33 ± 0.19** | **11.84 ± 0.18** | **9.92 ± 0.98** | **11.27 ± 0.88** |
| YYG | FFN | -2.08 ± 0.39 | -1.34 ± 0.12 | -2.64 ± 0.13 | -3.36 ± 0.18 |
| | B2F-MB | -3.84 ± 0.08 | -6.99 ± 0.56 | -8.84 ± 0.96 | -5.61 ± 0.27 |
| | B2F-NOGRAPH | -1.25 ± 0.70 | -0.7 ± 0.90 | -6.13 ± 0.08 | -5.31 ± 0.06 |
| | B2F-BSEQ | -1.78 ± 0.74 | -2.26 ± 0.83 | -0.79 ± 0.21 | -0.62 ± 0.27 |
| | Back2Future | **8.93 ± 0.26** | **6.32 ± 0.44** | **7.32 ± 0.42** | **5.73 ± 0.66** |
| UMASS-MB | FFN | -3.25 ± 0.38 | -5.74 ± 0.75 | -1.01 ± 0.18 | -5.28 ± 0.07 |
| | B2F-MB | -8.2 ± 0.61 | -7.54 ± 0.26 | -6.49 ± 0.29 | -7.56 ± 0.30 |
| | B2F-NOGRAPH | -2.16 ± 0.22 | -1.88 ± 0.67 | -2.15 ± 0.24 | -2.87 ± 0.34 |
| | B2F-BSEQ | 1.58 ± 0.49 | 0.86 ± 0.17 | 0.36 ± 0.06 | 0.96 ± 0.82 |
| | Back2Future | **5.43 ± 0.51** | **4.66 ± 0.63** | **3.32 ± 0.76** | **3.11 ± 0.29** |
| CMU-TS | FFN | -5.24 ± 0.57 | -4.93 ± 0.39 | -3.12 ± 0.71 | -0.65 ± 0.81 |
| | B2F-MB | -8.17 ± 0.34 | -8.21 ± 0.24 | -3.72 ± 0.32 | -6.11 ± 0.84 |
| | B2F-NOGRAPH | -0.67 ± 0.69 | -0.57 ± 0.32 | -0.46 ± 0.07 | -1.77 ± 0.79 |
| | B2F-BSEQ | 1.46 ± 0.33 | 1.05 ± 0.16 | 2.38 ± 0.43 | 2.26 ± 0.02 |
| | Back2Future | **7.5 ± 0.60** | **8.04 ± 0.58** | **5.73 ± 0.19** | **6.22 ± 0.58** |

**Baselines:** Due to the novel problem, there are no standard baselines. Therefore, our baselines are used to study the effect of using BSEQs and model bias(MB) on refinement. (a) FFN: train a simple feed-forward neural network for regression task that takes as inputs model's prediction and real-time target to predict the stable target. (b) B2F-MB: exploits only model bias and uses the MODELPREDENC architecture and append a linear layer that takes encodings from MODELPREDENC and model's prediction (c) B2F-BSEQ: exploits only BSEQ without model bias and only uses BSEQENC architecture and append a linear layer that takes encodings from BseqEnc and model's prediction (d) B2F-NOGRAPH: does not use BSEQ similarity from GRAPHGEN by removing graph convolutional layers and retain only RNNs.

**Refining real-time model-predictions:** We compare the mean percentage improvement (decrease) in scores of B2F refined predictions of diverse set of top models w.r.t stable targets over 50 US states. We observe that B2F is the only method, compared to baselines, that improves scores for *all* candidate models consistently (Table 2). The poor scores of baselines also shows the necessity of incorporating both backfill information (unlike FFN and B2F-MB) and model prediction history (unlike B2F-BSEQ and B2F-NOGRAPH).

We achieve substantial avg. improvements of 6.93% and 6.79% in MAE and MAPE respectively with low standard deviation across 29 test weeks which shows that the improvements were consistent across time and not just over few weeks that experienced large revision anomalies. Candidate models refined by B2F show improvement of over 10% in over 25 states and over 15% in 5 states (NJ, LA, GA, CT, MD). The improved predictions of CMU-TS and GT-DC (ranked 3rd and 4th in COVID-19

Forecast Hub) due to B2F, outperform all the models in the hub (except for ENSEMBLE) with 7.17% and 4.13% improvements in MAE respectively. UMASS-MB, ranked 2nd, is improved by 11.24%. B2F also improves ENSEMBLE, the current best-performing model of the hub, by 3.6% - 5.18% with over 5% improvement in 38 states, and with IL and TX experiencing over 15% improvement.

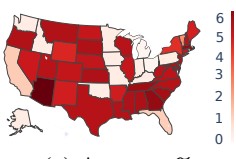

(a) Average %
decrease of MAE

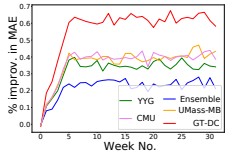

(b) % improve. in
MAE for each week

Figure 6: *(a)* B2F *refines* ENSEMBLE *predictions significantly for most states. (b) efficacy of* B2F *ramps up within 6 weeks of revision data.*

**Rectifying real-time model-evaluation:** We evaluate the efficacy of B2F in rectifying the real-time evaluation scores for the LBRP problem. We noted in Obs 5 that real-time MAE was lower than stable MAE by 9.6 on average. The difference between B2F rectified estimates MAE stable MAE was reduced to 4.4, a 51.7% decrease (Figure 7). This results in increased MAE scores across most regions towards stable estimates. Eg: We reduce the MAE difference in large highly populated states such as GA by 26.1% (from 22.52 to 16.64) and TX by 90% (from 10.8 to 1.04) causing an increase in MAE scores from real-time estimates by 5.88 and 9.4 respectively.

**Refinement as a function of data availability:** During the initial weeks of the pandemic, we have access to very little revision data both in terms of length and number of BSEQ. So we evaluate the mean performance improvement for each week across all regions (Figure 6b). B2F's performance ramps up and quickly stabilize in just 6 weeks. Since signals need around 4 weeks (Obs 2) to stabilize, this ramp-up time using small amount of revision data is impressive.

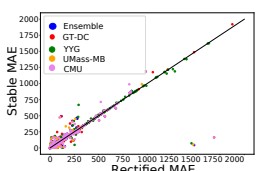

Figure 7: B2F *rectified MAE are closer to stable MAE*

**B2F adapts to anomalies** During real-time forecasting, models need to be also robust to rare events of large anomalous data revisions, like during the initial stages of the pandemic when data collection was not fully streamlined. Consider observation week 5 where there was an abnormally large revision to deaths nationwide (Figure 8a) when the BERR was 48%. B2F still provided significant improvements of up to 74.2% for most model predictions (Figure 8b).

**B2F refines GDP forecasts** To evaluate the extensibility of B2F to other domains that encounter the problem of backfill, we tested on the task of forecasting US National GDP using 25 macroeconomic indicators from past and their revision history for years 2000-2021. We found that B2F improves predictions of candidate models by 6%-15% and significantly outperforms baselines. The details of the dataset and results are found in Appendix Sections B, E.2.

## 5 CONCLUSION

We introduced the important and challenging multi-variate backfill problem using COVID-19 and GDP forecasting as examples. We compiled and released the comprehensive *CoVDS* dataset to study revision patterns in features and targets as well as aid in COVID-19 forecasting. We presented

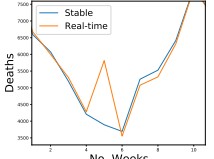

(a) Revision of US deaths
on week 5

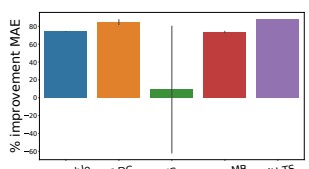

(b) % decrease in MAE due to B2F
refinement on week 5 targets

Figure 8: B2F *adapts to abnormally high revisions*

Back2Future (B2F), the novel deep-learning method to model this phenomenon, which exploits our observations of cross-signal similarities using Graph Recurrent Neural Networks to refine predictions and rectify evaluations for a wide range of models. Our extensive experiments showed that leveraging similarity among backfill patterns as well as model bias via our proposed method leads to significant 6 - 11% improvements in all the top models.

As future work, our work can potentially help improve data collection and alleviate systematic differences in reporting capabilities across regions. For example, B2F provides significant gains consistently across time including when there are large anomalies in data. Therefore, our revision modelling approach can be helpful for anomaly detection (Homayouni et al., 2021). We can also study how backfill can affect uncertainty calibration in time-series analysis (Yoon et al., 2020). Adapting to situations where data revisions can occur at different frequencies is another research direction.

## 6 ETHICS STATEMENT

The features used in the *CoVDS* dataset and for GDP forecasting are publicly available and anonymized without any sensitive information. Our backfill refinement framework and B2F is generalizable to any domain that deals with real-time prediction tasks with feature revisions. Due to the relevance of our dataset to public health and macroeconomics, prospects for misuse should not be discounted. The disparities in data collection across features and regions can also have implications on equity of prediction performance and is an interesting direction of research.

## 7 REPRODUCIBLITY STATEMENT

As described in Section 4, we evaluated our model over 5 runs with different random seeds to show the statistical significance of our method. We also provide a more extensive description of hyperparameters and data pre-processing in the Appendix. The code for B2F and the *CoVDS* dataset is publicly available at `https://github.com/AdityaLab/Back2Future`.

## 8 ACKNOWLEDGEMENTS

This work was supported in part by the NSF (Expeditions CCF-1918770, CAREER IIS-2028586, RAPID IIS-2027862, Medium IIS-1955883, Medium IIS-2106961, CCF-2115126), CDC MInD program, ORNL, and faculty research awards from Facebook, funds/computing resources from Georgia Tech.

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
