# OpenReview forum: "Back2Future: Leveraging Backfill Dynamics for Improving Real-time Predictions in Future"
_ICLR.cc/2022/Conference — ICLR 2022 Poster_

### Official Review · Reviewer_apgM · 2021-10-31

**Correctness:** 4
**Technical Novelty And Significance:** 3
**Empirical Novelty And Significance:** 3
**Recommendation:** 6
**Confidence:** 4

**Main Review:**

The paper proposes an approach to overcoming the so-called backfill problem when time-series data are revised to correct previous recorded values. The authors propose a recurrent neural network to carry out the task. The technical part of the paper is very well written and the intuition behind various modeling decisions is explained in a clear and simple way. I also appreciated the careful preliminary data analysis of the covid data that motivate and back the model proposal. The authors carry out a comparative evaluation against several CDC-approved forecasting models showing convincing better results.

Strengths:
- The paper is very well written. The problem is well presented, notation is defined carefully, and the model is described with enough detail.
- A careful COVID data analysis serves to motivate and back the model proposal.
- The work is very relevant to the current situation and ongoing research

Minor comments:
- is it possible to give some reasons for the revisions in the case of covid?
- In section 2.1, clarify what is the spatial analysis unit adopted.
- What is the model performance when a finer spatial unit is adopted such as counties.



**Summary Of The Paper:**

Summary: The paper proposes a real-time forecasting model for situations where the predictive features and the response variable are both subject to recording delays. The authors propose a recurrent neural network model and evaluate it against several baseline models using a COVID dataset.

**Summary Of The Review:**

This is a very well-written paper with good explanations, proper motivation, clear objectives, and results.

---

> ### Author Response · Authors · 2021-11-19
> **Response to Reviewer apgM**
>
> We thank the reviewer for their valuable feedback and encouraging comments.
>
> ### Regarding technical novelty:
> We just wanted to briefly reemphasize that our work is the first to provide a general unifying framework to tackle the general multivariate data revision problem (in features as well as targets) for real-time time-series prediction, which has not been studied in the literature before. We are also the first to introduce a representation learning framework to model the backfill dynamics by:
>
> 1. Leveraging Graph Neural Networks with recurrent modules to learn from sparse relations across features based on similar revision patterns.
> 2. Introducing a two-step training routine that uses self-supervised pre-training and later efficiently fine-tuned to refine any candidate model.
>
> Leveraging these innovations, B2F significantly improves real-time predictions of the different state-of-art models in real-time forecasting on the important problems of Covid-19 and economic GDP forecasting.
>
> ## Is it possible to give some reasons for the revisions in the case of covid?
>
> We briefly mentioned the various general causes for backfill from past work in the second paragraph of Section 1. Most of these apply to the Covid-19 pandemic as well. However, identifying the exact cause of reporting delays and revisions for a specific ongoing pandemic like Covid-19 is challenging and an open problem. Some causes which have been found by other groups include natural disasters (such as the Texas winter storms during Feb 2021), man-made factors such as public-health agencies changing reporting methodologies as well as weekly and seasonal biases in reporting [1, 2]. It is an interesting idea if correction methods like B2F can also aid in this ‘attribution’ task.
>
>
> ## In section 2.1, clarify what is the spatial analysis unit adopted.
>
> We collected the features for each of the 50 US states (first paragraph of Section 2.1). We will state this more clearly in the final version.
>
> ## What is the model performance when a finer spatial unit is adopted such as counties.
>
> This is an interesting question. Unfortunately, at present, collecting a good set of features and their revisions at lower levels (like all ~3K US counties) is challenging, and several of the features are not even available. However, given that B2F has been designed for general multivariate features leveraging model revision patterns, it will be interesting to explore this aspect. We will plan to study this in future work.
>
> [1] Rodriguez, Alexander, et al. "Deepcovid: An operational deep learning-driven framework
>      for explainable real-time covid-19 forecasting." AAAI (2021).
>
> [2] Pollett, Simon, et al. "Identification and evaluation of epidemic prediction and forecasting
>      reporting guidelines: A systematic review and a call for action." Epidemics 33 (2020):
>      100400.

---

### Official Review · Reviewer_9pAe · 2021-11-02

**Correctness:** 4
**Technical Novelty And Significance:** 4
**Empirical Novelty And Significance:** 4
**Recommendation:** 8
**Confidence:** 4

**Main Review:**

Strengths:

- Characteristics of the backfill dynamics problem is very well motivated with observations from COVID-19 data that the authors have collected. While people have long known that corrections to data over time will have effects on forecasts, this paper is the first (to my knowledge) that carefully examines the prevalence and effects of backfill dynamics.
- Very general B2F pipeline that can be applied to "post-process" the predictions of any forecasting model. The wide applicability could lead to high impact.
- Strong experimental results on real data, showing consistent improvement of COVID-19 death forecasts for 5 different candidate models.
- Paper is very well organized and written, with only some minor issues that I list below.

Weaknesses:

- Fit for ICLR may not be the best. There's not really a contribution towards learning representations, as the B2F pipeline integrates existing neural network architectures. I think this is a better fit for a data mining venue such as KDD.
- Text in many figures, including Figure 1, 3, and 4, is too small to be readable. I recommend removing some of the results, for example, converting Table 2 to a figure that shows only the ensemble model. A more complete version of Table 2 is available in Table 4 of the appendix anyways.

Minor presentation issues:

- Figure 3 caption: Usually an x-y plot is described as y vs. x, whereas this caption is x vs. y.
- Obs. 4: non-monotonous $\rightarrow$ non-monotonic
- Page 9, 2nd paragraph: "B2F rectified estimates MAE stable MAE": Insert "in" between "MAE" and "stable".

**Summary Of The Paper:**

The authors consider the effects of backfill dynamics--the correction of historical data--on time series prediction. They use COVID-19 forecasting as the motivating application. Both the features (ER visits, hospitalization rate, etc.) and the target (deaths) are subject to revision, and the authors collect a data set showing that the backfill error due to revisions can be extremely large. The authors make several other interesting observations regarding backfill error and sequences in their data. A key finding is that real-time forecasts tend to overestimate their accuracy when compared to the stable accuracy after errors are corrected. The authors then propose a Back2Future (B2F) pipeline that can be used to refine predictions from a model when backfill dynamics are present. They demonstrate consistent improvement on stable accuracy for multiple COVID-19 forecasts.

**Summary Of The Review:**

The paper contributes both new data and a new problem setting, motivated by forecasting COVID-19 deaths over time. Due to the high novelty and very general proposed B2F pipeline for refining model predictions, this paper could have high impact, although it may not be the best fit for the ICLR audience.

---

> ### Author Response · Authors · 2021-11-19
> **Response to Reviewer 9pAe**
>
> We thank the reviewer for their valuable feedback and encouraging comments. We are excited about this paper which is the first to introduce a representation learning framework to model backfill dynamics and connect it with the general ML problem of real-time forecasting. We also thank the reviewer for their suggestions regarding improving the presentation. We will improve the font size of the figures and fix the corrections pointed to by the reviewer in the final version.

---

> > ### Comment · Reviewer_9pAe · 2021-11-29
> > **I support the paper**
> >
> > After reading over the other reviews and author responses, I continue to support the paper and am leaving my score unchanged.

---

### Official Review · Reviewer_4Gyj · 2021-11-11

**Correctness:** 4
**Technical Novelty And Significance:** 4
**Empirical Novelty And Significance:** 4
**Recommendation:** 8
**Confidence:** 4

**Main Review:**

The authors propose an pipeline to alleviate the effects of revising historical data
(backfill problem) in the context of timeseries prediction.

Strong points:
- Well motivated. The authors present clearly the backfill problem and its effects on the predictions thus enabling the reader to understand the importance of the suggested solution.
- Well written. The use of the COVID dataset as the driving
application has successfully contributed to the understanding of the pipeline.
- Generalizability. The pipeline can be deployed to refine the predictions and improve the performance of any model.
- Thorough experimental analysis and strong results.
- Sound technical backbone.

Weak points:
- At parts the writing feels aesthetically dense. I would suggest that the authors try and cut some of
the text so that tables/figures such as Table 2 (which is one of the main results of the paper)
is not so tightly surrounded by text.

-- Other:

Abstract
"Back2Future, that aims to refines a given model’s predictions ..." -> "Back2Future, that aims to refine a given model’s predictions ..."

Introduction
Worth unfolding the CDC acronym to centers for disease control and prevention. Currently I suspect that the majority of the readers will be well aware of this acronym but this might not be the case as some point later in the future.

" many times after and is known as the ’backfill’ phenomenon.The various factors ..." --> please mind the right direction in the single quotes and for aesthetics, it is worth leaving a space after a period and before the beginning of the next sentence.

’backcasting’ --> please correct the direction of the single quotes.

"Building on our empirical observations, we formulate the problem BFRP, ..."--> Please unfold he acronym as this is the first time introduced.

Nature of backfill dynamics
"Also, note that the data in the backfill is the same used for real-time forecasting, but just seen from a different perspective." --> This needs clarification. What is the different perspective.


**Summary Of The Paper:**

The authors deal with the problem of revising previous recorded data and its effect on timeseries predictions. They showcase how revisions in past data, quantified as the backfill error, can introduce a considerable error in predictions. Towards that, they propose a novel deep learning approach, the Back2Future, that refines the model predictions using backfill dynamics. They demonstrate its efficiency on a real COVID dataset.


**Summary Of The Review:**

The proposed work is well motivated, well written, sound and contributes a model that solves a very interesting problem in timeseries prediction with strong results. Its capability of use along with any model adds to its value. I would be happy to see this work accepted.

---

> ### Author Response · Authors · 2021-11-19
> **Response to Reviewer 4Gyj**
>
> We thank the reviewer for their valuable feedback and encouraging comments. We will go through the paper again and try to declutter dense sentences. We will also make the suggested corrections to the typos pointed to by the reviewer.

---

### Decision · Program_Chairs · 2022-01-20

**Decision:**

Accept (Poster)

**Comment:**

This paper introduces Back2Future, a deep learning approach for refining predictions when
backfill dynamics are present.

All reviewers agree on that the authors successfully motivate their work and
introduce a topic of great interest, i.e. that of dealing with the effect of revising previously recorded data and its effect
timeseries predictions. The reviewers also underline the strong and thorough experimental section.
Among the reviews is also underlined the potential impact of the work for the research domain.

Many thanks to the authors for replying to the minor concerns raised.

I concur with the reviews and find this submission very interesting, convincing and thus
recommend for accept.

Thank you for submitting the paper to ICLR.